# The Significance of Groundwater Flow Modeling Study for Simulation of Opencast Mine Dewatering, Flooding, and the Environmental Impact

**Jacek Szczepiński**

Poltegor-Institute, Institute of Opencast Mining, 51-616 Wrocław, Poland; jacek.szczepinski@igo.wroc.pl

**Abstract:** Simulations of open pit mines dewatering, their flooding, and environmental impact assessment are performed using groundwater flow models. They must take into consideration both regional groundwater conditions and the specificity of mine dewatering operations. This method has been used to a great extent in Polish opencast mines since the 1970s. However, the use of numerical models in mining hydrogeology has certain limitations resulting from existing uncertainties as to the assumed hydrogeological parameters and boundary conditions. They include shortcomings in the identification of hydrogeological conditions, cyclic changes of precipitation and evaporation, changes resulting from land management due to mining activity, changes in mining work schedules, and post-mining void flooding. Even though groundwater flow models used in mining hydrogeology have numerous limitations, they still provide the most comprehensive information concerning the mine dewatering and flooding processes and their influence on the environment. However, they will always require periodical verification based on new information on the actual response of the aquifer system to the mine drainage and the actual climate conditions, as well as up-to-date schedules of deposit extraction and mine closure.

**Keywords:** open pit; dewatering; flooding; groundwater modeling; Poland

## 1. Introduction

In the mining industry, water-related problems are among the most important aspects which can decide whether a new mine development will be reasonable or even feasible. For an assessment into the costs of mining operations, the rate of mine water inflow, dewatering technology, and the environmental impact of mine drainage are all important factors. In the past, one would mainly focus on the hazards connected with mine water inflow; at present, most of the attention is focused on flooding the post-mining voids and the environmental impact assessment [1].

At the initial stage of research, to estimate the mine water inflow, one typically applies the methods of hydrogeological analogy or hydrogeological balance. Hydrogeological calculations are usually performed using classical analytical methods described in numerous textbooks [2–5]. Further investigations are dominated by numerical methods. They are used throughout the mine's whole life-cycle. Numerical methods allow forecasting the process of mine dewatering as well as the process of flooding the post-mining excavations with greater accuracy than any other method. They can assess the impact of dewatering on groundwater, surface water, water chemistry, water intakes, soil, flora, farmlands and forests, land subsidence, and others. Particularly, the process of flooding post-mining voids with complex geometry and diversified hydrogeological conditions must rely on numerical models [6].

Modeling methods were widely used in mining hydrogeology since the beginnings of their application. Nevertheless, the groundwater flow modeling applied in the mining industry belongs to

the most demanding methods in hydrogeology. Because mine dewatering and groundwater rebound are changing in time and in space, they should be solved in transient conditions. In the area covered by the dewatering impact, both boundary conditions and hydrogeological parameters need to be updated in response to hydrogeology and hydrology conditions as well as scope and schedule of mining works changing due to mining activities.

The topics related to modeling study on mine dewatering and its environmental impact, including groundwater/surface-water interactions, groundwater resources, and hydrogeochemistry, have been the subject of many scientific publications which are mentioned in this paper. Papers on mine water and its impact on the environment appear mainly in the International Mine Water Association post-conference proceedings and in scientific journals, especially Mine Water and the Environment [7–9], Groundwater [10–12], Environmental Geology [13,14], and others [15]. In addition, the following textbooks deserve special attention: Mine Water Hydrology, Pollution, Remediation (in which mine water hydrology and chemistry are presented with an explanation of the complexities of mine water pollution and the hydrogeological context of its formation) [16], and Construction Dewatering and Groundwater Control (which presents the theory and practice of dewatering from the engineering point of view) [5]. In recent years, extensive monographs on the water management related to flooding of post-mining voids have appeared: Water Management at Abandoned Underground Flooded Mines. Fundamentals, Tracer Tests, Modeling, Water Treatment [17]; Mine Pit Lakes: Characteristics, Predictive Modeling, and Sustainability (Management Technologies for Metal Mining Influenced Water) [18]; and Acidic Pit Lakes. The Legacy of Coal and Metal Surface Mine [19].

The aim of the paper is to present the specificity of the application of groundwater flow modeling for simulation of opencast mine dewatering, flooding, and the environmental impact. This paper is a result of many years of the author's experience on groundwater flow modeling for open-pit dewatering and its flooding as well as any environmental impact caused by mine dewatering in Poland and abroad.

## 2. Application of Numerical Modeling in Mining Hydrogeology

The electric analogue computer model using the method of electrohydrodynamic analogy appeared in the 1930s in the oil industry in the United States. Numerical models in calculations of groundwater flow model have been used since the 1960s. In 1965, a paper was published on the use of these methods for solving the groundwater flow equation [20], and, in 1968, a digital-computer program was written to solve linear, parabolic, and partial-differential equations based upon an implicit finite-difference technique [21].

A significant increase in the use of numerical methods occurred in the first half of the 1970s, with the appearance of the Prickett-Lonnquist Aquifer Simulation Model (PLASM) [22] and the U.S. Geological Survey (USGS) model [23], which solved the three-dimensional groundwater flow model based on the finite difference method. The history of the beginnings of groundwater flow modeling was extensively presented by Bredehoeft [24].

In the mining industry, groundwater flow models are applied in a wide range using both the finite difference method and the finite element method for porous aquifers as well as fissured aquifers. Currently, both in Poland and around the world, the MODFLOW program based on the finite difference method [25,26] and the FEFLOW program based on the finite element method [27] are used the most extensively in mining hydrogeology. The modeling results usually simultaneously include a prediction of the mine water inflow, an input parameters into an environmental impact assessment, and the water management of the post-mining excavation. The present state of groundwater flow modeling applications in mining hydrogeology was presented by Rapantova et al. [9].

In mining, groundwater flow modeling represents both regional (aquifer system) and local issues (mining excavation area). Numerical modeling is used during each stage of the mining operation, beginning with deposit exploration to mine decommissioning with mine site rehabilitation. Its purpose is to deliver reliable forecasts of mine water inflow and the input parameters into an environmental impact assessment of mine drainage with reference to different scenarios of the deposit opening up and



its extraction, as well as rehabilitation of post-mining excavation. The results of numerical simulations enable to provide reliable data which help with the decision making at each stage of mining operation:

- Stage of identification, documentation, and opening up of the deposit: Assessment of hydrogeological conditions of the deposit area, mine water inflow calculation, suggestion on dewatering technology, determination of the cone of depression and its boundaries, impact assessment on surface water (watercourses and reservoirs) and groundwater in each dewatering layer, determination of the area of environmental impact of mining, and indication the necessary measures to minimize the negative impact on the environment.
- Stage of deposit exploitation: Forecasting the mine water inflow intensity and the change of groundwater level resulting in changes of geotechnical conditions of slopes in time and space as the mine develops, as well as determining impacts on the environment based on real data gathered during previous dewatering.
- Stage of mine decommissioning: An indication of a rational way of flooding and management the post-mining void, determination of time span for filling an excavation with water, final water level in the post-mining reservoir, and the forecasting of quantitative and qualitative changes in the reservoir and its environment during and after flooding.

In many cases, models are also accomplished in order to confirm or reject the mine drainage impact on the environment. For example, the results of numerical simulation can provide the proof on real impact of mine dewatering and/or climatic conditions on groundwater level and groundwater resources.

## 3. Specificity of Application the Modeling Methods for Opencast Mining

The procedure for developing groundwater flow models in surface mining conditions generally complies with the overall modeling methodology which is presented in many textbooks and guidelines [28–31]. However, it must be adjusted to the specificity of open pit mine drainage operations.

### 3.1. Groundwater Flow System Representation

Groundwater flow in the vicinity of open pits or underground mines is three-dimensional in most cases; consequently, 3-D numerical groundwater flow models must be based on 3-D hydrogeological data [32] and should be simulated by a full three-dimensional model, with the hydraulic head simulated for each model layer. It is particularly important in the areas adjacent to the open pits, where the groundwater flow has a substantial and important 3-D component. For an assessment of the range and extent of the cone of depression or the mine water inflow, one can apply a quasi-three-dimensional model which only represents the vertical flow through semi-permeable layers (Figure 1) [33].

A simulation of mine drainage and its influence on the water environment needs a huge amount of data—more than for other regional models. In order to develop reliable forecasts, it is necessary to recognize the hydrogeological and hydrological conditions of the deposit and neighborhood aquifers as well as to identify all environmental, mining and technological factors which can affect the mine water inflow. The most important factors are: hydrogeological parameters, dewatering technology, the recharge of aquifers from precipitation and surface water as well as management of post-mining excavations.

The conceptual model requires not only detailed the identification of hydrogeological conditions within of the model area but also a correct representation of the mine drainage system (Section 3.2). The watercourses, lakes, and reservoirs within the model area should be simulated, taking into consideration their possible drying as well as the re-wetting in the case of groundwater rebound. General principles of accepting internal and external boundary conditions have been presented extensively in many studies [28,34–37].

An important step in modeling study is the development of a conceptual model of recharge processes. It is formed by integrating many factors: Spatial and temporal variability in recharge, climate, soil and geology, surface topography, hydrology, depth to water, and others [38]. The term "potential

recharge" was introduced by Rushton [39]. This type includes the excesses of precipitation over evapotranspiration, which subsequently disappear through a local discharge system or by evaporation from the saturated zone, but which could become a "permanent" recharge by the lowering of a shallow water table after extraction. Moreover, lowering a shallow water table can induce additional recharge by reducing evapotranspiration. This concept of potential recharge is important for modeling future conditions [40]. In Poland, the groundwater table in pre-mining condition is frequently 1–3 m below the land surface elevation, so evapotranspiration is a very important factor in groundwater balance. In mining areas, due to the extinction of evapotranspiration from groundwater as a result of a groundwater table lowering (as well as evaporation from land surface and zone of aeration), induced groundwater resources (part of a groundwater renewable resources) are created in the area of the cone of depression [41].

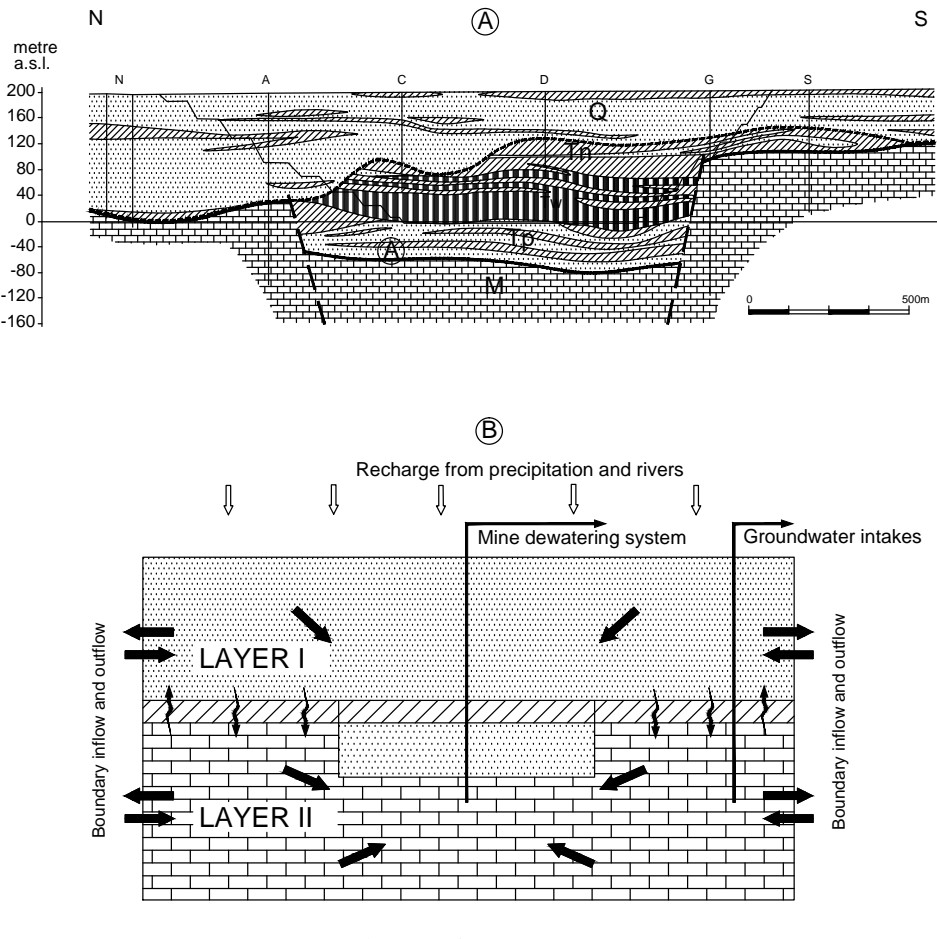

**Figure 1.** Simplified geological cross-section N-S with the conceptual model suitable for numerical modelling. Explanation: **1**—porous, permeable formations; **2**—impermeable and slightly permeable formations; **3**—fissured-karstic formations; **4**—lignite; **5**—boundary of the Quaternary aquifer; **6**—faults; **7**—symbols of aquifers; **8**—open-pit limits; **9**—dewatering wells barriers; **10**— direction of groundwater flow; **11**—direction of groundwater percolation [33].

Recharge and evaporation boundary conditions should ensure the appropriate representation of effective infiltration and its changes during water table fluctuation whenever such changes take place. Generally, recharge in groundwater models can be represented with constant head boundary conditions, specified-flux boundary conditions, mixed-boundary conditions, or some combination of the mixed and flux boundary conditions. Modeling studies where transient drawdown cones are developing at pumping centers are an example where a specified-flux boundary is appropriate for recharge estimation.

However, for variably controlled recharge, the modeler can invoke mixed-boundary conditions or some combination of the mixed and flux boundary conditions [42].

### 3.2. Simulation of Opencast Mine Dewatering System

The opencast mine dewatering system is modeled on the basis of information of the planned location, scope, and schedule of mining works related to the deposit opening up, the mining face advancement, and the development of the internal dump. Before starting the simulation, it should be determined at what time and to what depth the groundwater level should be lowered, ensuring the effective deposit dewatering and safety of slopes of the active mine or post-mining voids. For this purpose, it is necessary to obtain information from the top and bottom of the deposit seam as well as the forecasted bench levels. Guidelines for mine drainage are developed on the basis of assumptions regarding the time advance with which the overburden should be dewatered prior to the start of mining works. Under Polish law, it is assumed that the required lowering of the groundwater table (usually one meter below the bottom of the pit) should be obtained for one year before the deposit begins to be exploited. This requires gradual lowering of pressure in subsequent years.

The location of the mine drainage system and the required groundwater level change, both in time and in space, must be performed in transient conditions. It can be simulated by boundary conditions of Type I: Hydraulic head as a function of time; Type II: Flux as a function of time; or Type III: Flux as a function of head. In the case of semi-permeable layers between aquifers, boundary conditions should be assigned separately for each layer on the model. The hydraulic contacts between all water-bearing layers allow for the simulation of the drainage system by the boundary condition assigned to the lowest aquifer, usually the most important from a dewatering point of view. In MODFLOW, for a mine dewatering system simulation, the Time-Variant Specified-Head Package can be used, which allows the user to specify head boundaries that can change within or between stress periods [43].

### 3.3. Simulation of Post Mining Voids Flooding

To develop a reliable conceptual model for flooding of the post-mining void, the spatial parameters of a final excavation are required (Figure 2). As a result of changing hydraulic and spatial parameters of aquifers and new stresses, which appear in the modeled area, the conceptual model used for pre-mining conditions may require improvement or even updating that represents mining or postmining conditions properly and improve the predictive numerical model.

Surface water reservoirs are usually hydraulically connected to the groundwater system and can play a significant role in it. In this sense, their impact is similar to watercourses. They can discharge or recharge groundwater. Most often in groundwater models, reservoirs are simulated like rivers and represented by boundary conditions of the first or third type [45,46]. In the conditions of water filling of post-mining excavations, the water level in the reservoir in the specified time intervals is not known. Therefore, calculation methods must be used in which the water table level in the reservoir will be part of the numerical groundwater flow model solution. Such an approach is necessary, for example, when assessing changes in groundwater flow conditions in the area of reservoirs, which are under the influence of artificial drainage (mining drainage, groundwater intakes), groundwater recharge (irrigation, recharge wells, flooding), and climate change [47].

One of the ways of simulating the post-mining reservoir is the so-called method of "high conductivity cells", which consists in assigning to the cells representing the reservoir a high value of hydraulic conductivity, usually several orders higher than the hydraulic conductivity of adjacent aquifers—for example k = 10,000 m/d [47–50]. In the case of a transient simulation, it is necessary to assign a specific yield of 1.0 [49,51]. Cells representing the reservoir on the model then become part of the solution of the numerical equation of groundwater flow, which makes it possible to calculate the surface water table level (Figures 2 and 3).

The limitations regarding this method include, among others: 1: The method is limited to seepage lakes; 2: Changes in lake surface area are not accommodated without additional programming; 3: The

method may require a large number of iterations to converge; 4: The magnitude of the calculated head differential across the lake should be close to zero, but it is sensitive to the imposed regional gradient and the ratio of the hydraulic conductivity of the aquifer to the hydraulic conductivity assigned to the lake nodes [47].

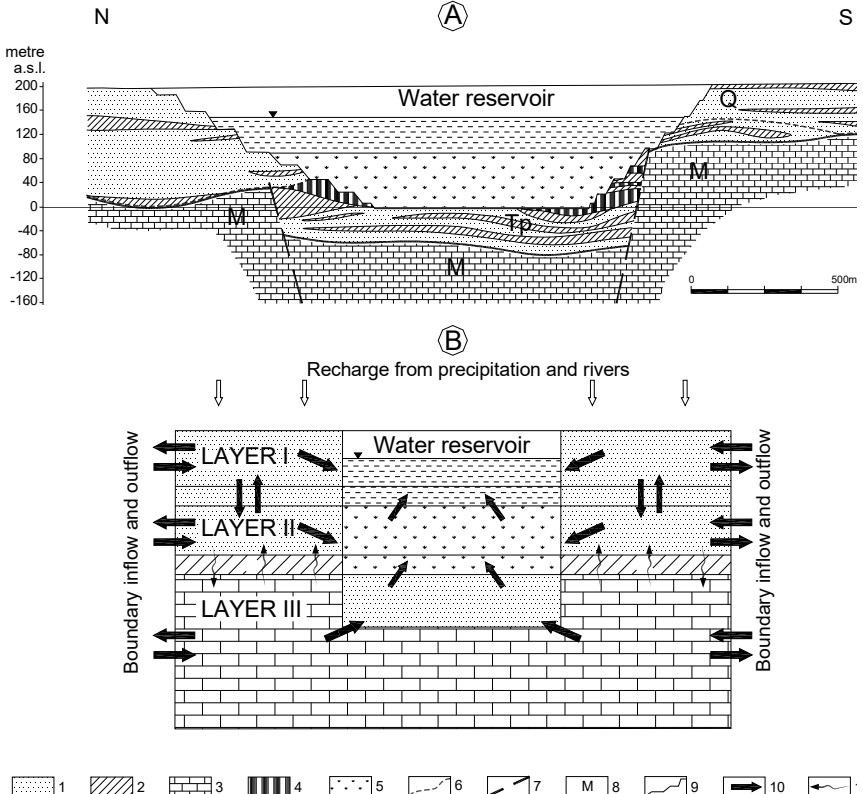

**Figure 2.** Simplified geological cross-section through the post-mining void (**A**) with the conceptual model suitable for numerical modeling (**B**). Explanation: **1**—porous, permeable formations; **2**—impermeable and slightly permeable formations; **3**—fissured-karstic formations; **4**—lignite; **5**—internal dump; **6**— boundary of the Quaternary aquifer; **7**—faults; **8**—symbols of aquifers; **9**—open-pit limits; **10**—direction of groundwater flow; **11**—direction of groundwater percolation [44].

### 3.4. Model Calibration and Verification

As mine drainage is a process variable in time and space, the models used in mining hydrogeology are solved in a transient simulation. However, at the first stage—i.e., for natural conditions, prior to the deposit dewatering—the model should be solved in steady-state conditions based on the average precipitation and evaporation data, hydrological data (flows and water levels in rivers and lakes), and hydrogeological data (groundwater levels and groundwater flows). Calibration performed under steady-state conditions enables the preliminary determination of aquifer recharge as well the horizontal and vertical hydraulic conductivity of aquifers.

In case we have data from pumping/aquifer tests or the first period of mine dewatering, a calibration to transient conditions can be performed to obtain the preliminary information on the specific yield and specific storage for the modeled area. In case of no such a data, the next steps of modeling in transient conditions must be based on archival data. After dewatering operation starts, model verification should proceed in transient conditions, taking into account the periods of mine dewatering. This requires at least one-time data. This enables the model users to improve the parameter-value estimates determined in the steady-state calibration as well as the calibration to the transient condition (if available) to improve the estimated values of the hydrogeological parameters (Figure 4).

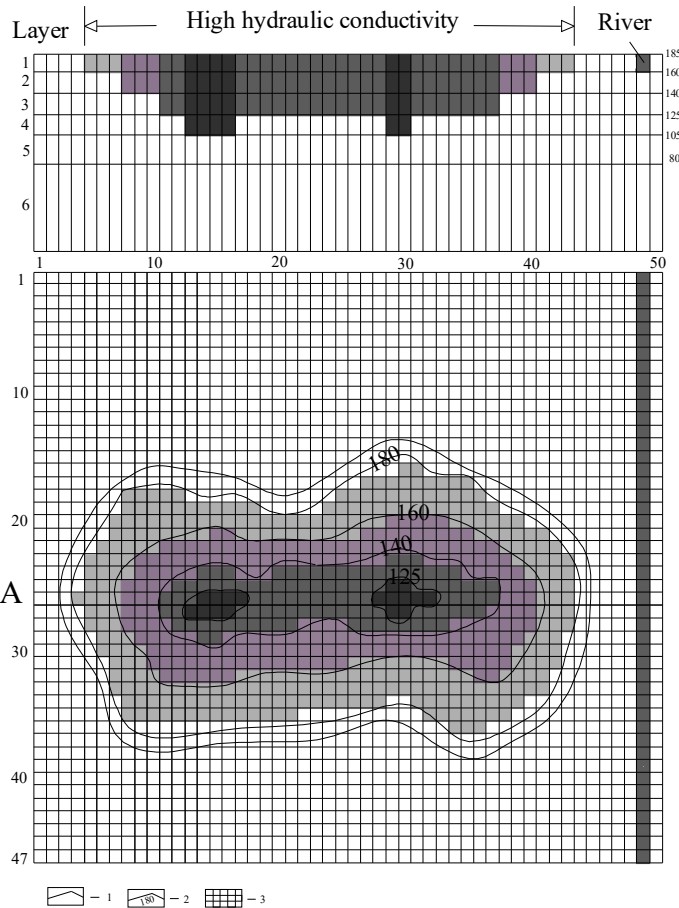

**Figure 3.** Simulation of a post-mining void using "high conductivity cell" method. Explanation: **1**—mine boundary area, **2**—surface contour interval, **3**—mesh discretization [33].

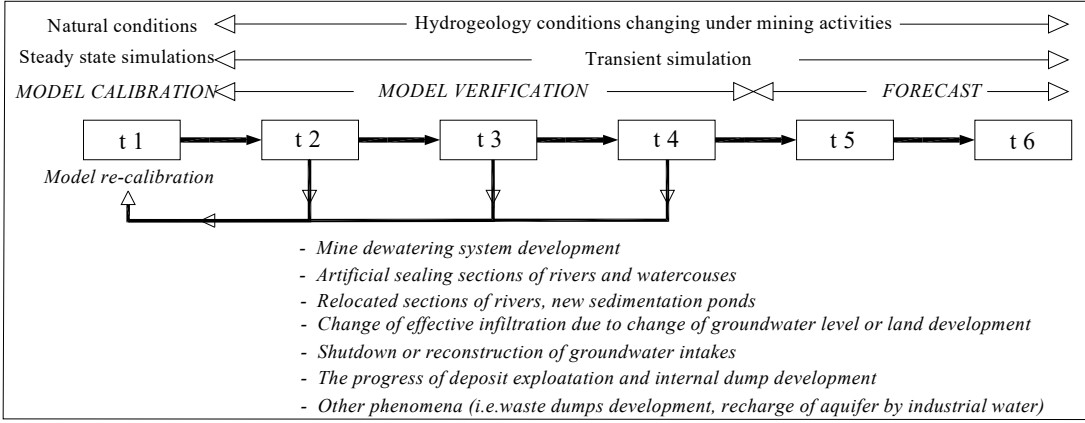

**Figure 4.** Calibration and verification of numerical models in the opencast mining activity area. Explanation: **t1–t6**: Stress periods; **t1**: Natural conditions (before dewatering operation); **t2–t4**: Dewatering period with real monitoring data; **t5–t6**: A forecast of mine dewatering and flooding of the post-mining void.

Numerical modeling should be understood as an iterative process in which reasonableness of results at each step should be checked before proceeding to the next step [29]. If the previously calibrated model does not reflect the real system reaction to the introduced stresses, the values assumed in the model should be re-calibrated using different datasets. The factors in need of possible updating include the boundary conditions, aquifer and aquitards, and hydrogeological and spatial parameters,

as well as a modification of the role of faults under the influence of deep mine drainage and leftover rock displacements due to mining activity.

In the mining area, boundary conditions representing development of the mine dewatering system should be assigned. Additionally, outside the mining area, it is necessary to update the boundary conditions to simulate artificially sealed or relocated rivers and watercourses, new constructed ditches and reservoirs, new water intakes, new land developments including industrial and municipal waste dumps, and a change of effective infiltration due to a lowering or increase of the groundwater table or an additional recharge of aquifers by industrial water from: Water-based drilling mud used during drilling activity, damaged sewerage and water systems, and landfills (if they exist). To quantify an uncertainty in the calibrated model caused by uncertainty in the estimates of aquifer parameters, stresses, and boundary conditions, a sensitivity analysis must be performed [28]. As a result, the water balance should be presented for steady-state simulations as well as for each stage of model verification. Its correctness should be evaluated by selected calibration criteria. The results obtained during calibration and verification of the model should specify:

- Hydro-structural system and aquifers transmissivity,
- specific yield and storage capacity of aquifers,
- changes in the leakage of water through semi-permeable layers,
- recharge of aquifers and its changes due to fluctuations in the groundwater table,
- sections of watercourses with decreasing or increasing flow.

At the stage of a forecasted simulation, a further updating of certain boundary conditions and hydrogeological or spatial parameters of the model can be required, which should be properly assigned to represent new stresses on the model (mine dewatering system development, new intakes, ditches, and ponds) and parameters representing an open pit area with the extracted deposit, the internal overburden dump development, or the post-mining excavations.

*3.5. Documenting of Modeling Study for the Mining Area*

Documenting the results of a modeling study in mining hydrogeology is more extensive compared to models used for groundwater-resource assessment. They must account for all boundary conditions and aquifers parameters subject to changes in time. While applying a model, the natural conditions, the dewatering period, and the post-mining excavations management should be taken into consideration. The results thus obtained, including the rate of mine water inflow, the hydraulic head in each modeled layer, the range of the cones of depression for all layers, the groundwater balance, and the hydrogeological conditions in each stage of mining operations and after its completion—ending with post-mining area reclamation—should be presented in both descriptive and graphical form for all the periods assumed. The sensitivity analysis for the model should be presented.

## 4. Problem of Aquifer Drying/Rewetting and its Representation in the Model

The drying/rewetting problem can be particularly damaging when undertaking groundwater modeling for mining applications [52]. In the case of confined layers where, due to dewatering, the thickness of the saturated zone does not change significantly in relation to its total thickness, the water-bearing layer may only be simulated by a transmissivity parameter. However, for a water-table aquifer and a confined layer that can be "dried" due to dewatering, it is necessary to specify on a model their top and bottom. The model may then face the problem of instability (lack of convergence) associated with the necessity of re-saturation of "dry cells" during the groundwater table rebound. In addition, due to groundwater table lowering beneath the bottom of a layer, the cells representing this layer are inactive during a calculation, and its recharge may be impossible, which is often inconsistent with the adopted conceptual model.

The innovative approach was only proposed as part of the block-centered flow MODFLOW-SURFACT package (BCF4), where, taking into account all the changes introduced in previous versions,

the pseudo-soil water retention automatically generated during the calculations was added to the formula for three-dimensional groundwater flow in a variably saturated system [53]. It does not allow for the appearance of inactive cells in areas where the water table has fallen below the bottom of the aquifer. The operation of this package consists in calculating, in each of the "dry" cells, the hydraulic height necessary to start water seepage through the unsaturated zone and not allowing the cells to be deactivated. Another solution to prevent complete drying of cells was presented by Doherty [52]. He introduced a modification to the block-centered flow package, so-called the "residual saturated thickness," allowing the reduced water conductivity of the aquifer to be maintained even if the groundwater table falls below the base of the cell. This modification allows leaving the cell active for groundwater flow.

Aquifer drying/rewetting process has a particular importance in case of deep open pit mines in which, for safety production, there are necessary dewatering many layers which are above the deposit. In a consequence of modeling study, the groundwater table on a model is lowering beneath the bottom of layers and the cells are inactive during a calculation (no groundwater flow). After the exploitation is ceased, the process of the groundwater table rebound starts, and the cells representing inactive layers have to be re-activated (re-saturated).

## 5. Use of Modeling Methods in Polish Opencast Mines

In Poland, groundwater flow numerical models have been used since the mid-1970s. The modeling studies for simulation of opencast mine dewatering were performed using the HYDRYLIB [54] program, which was based on the finite difference method and the FKWH [55] program, which was based on the finite element method. Currently, MODFLOW is the most popular. The numerical models are performed for sulphur and raw materials opencast mines. However, the most extensive groundwater flow models are used for the existing lignite opencast mine [56,57] (Figures 5 and 6) and proposed lignite opencast mines [58,59] (Figures 7 and 8).

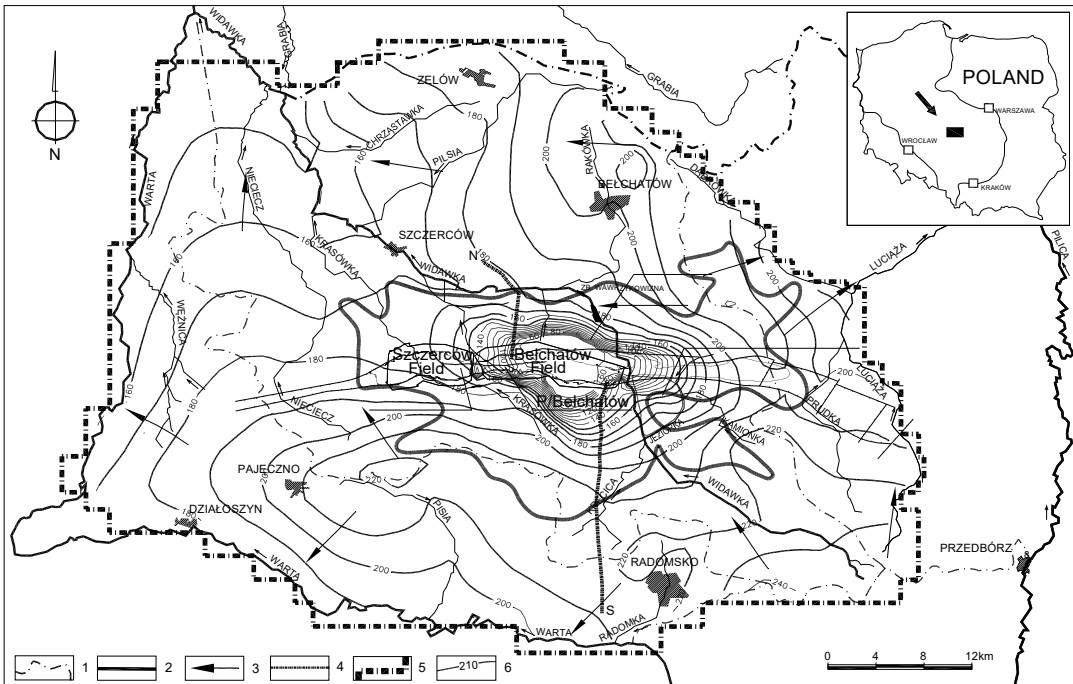

**Figure 5.** The range of the numerical model in the area of the Bełchatów lignite mine with groundwater contours for mining conditions. Explanations: **1**—watersheds; **2**—cone of depression range; **3**—groundwater flow direction; **4**—hydrogeological cross-section; **5**—model boundary; **6**—groundwater level in mining conditions [56].

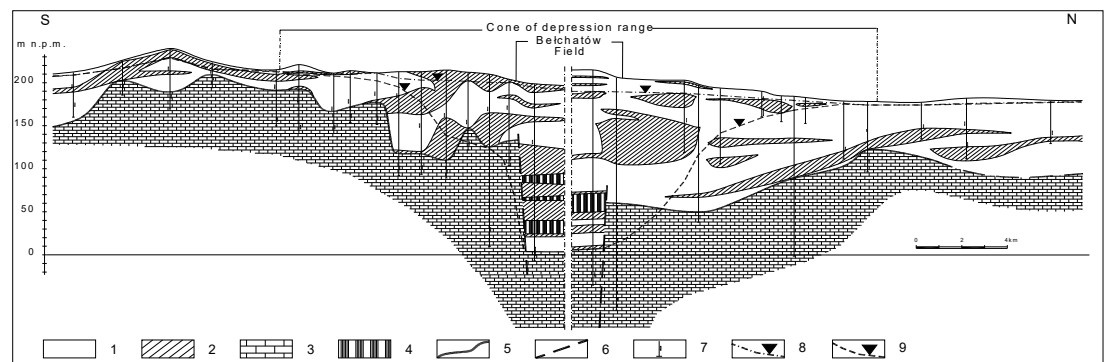

**Figure 6.** Hydrogeological cross-section through Bełchatów Opencast Mine. Explanations: **1**—fine sands; **2**—glacial tills; **3**—limestones, marls and dolomites; **4**—lignite; **5**—stratigraphic boundary; **6**—faults; **7**—screened intervals of a weeks; **8**—ground water level in pre-mining conditions; **9**—ground water level in mining conditions [57].

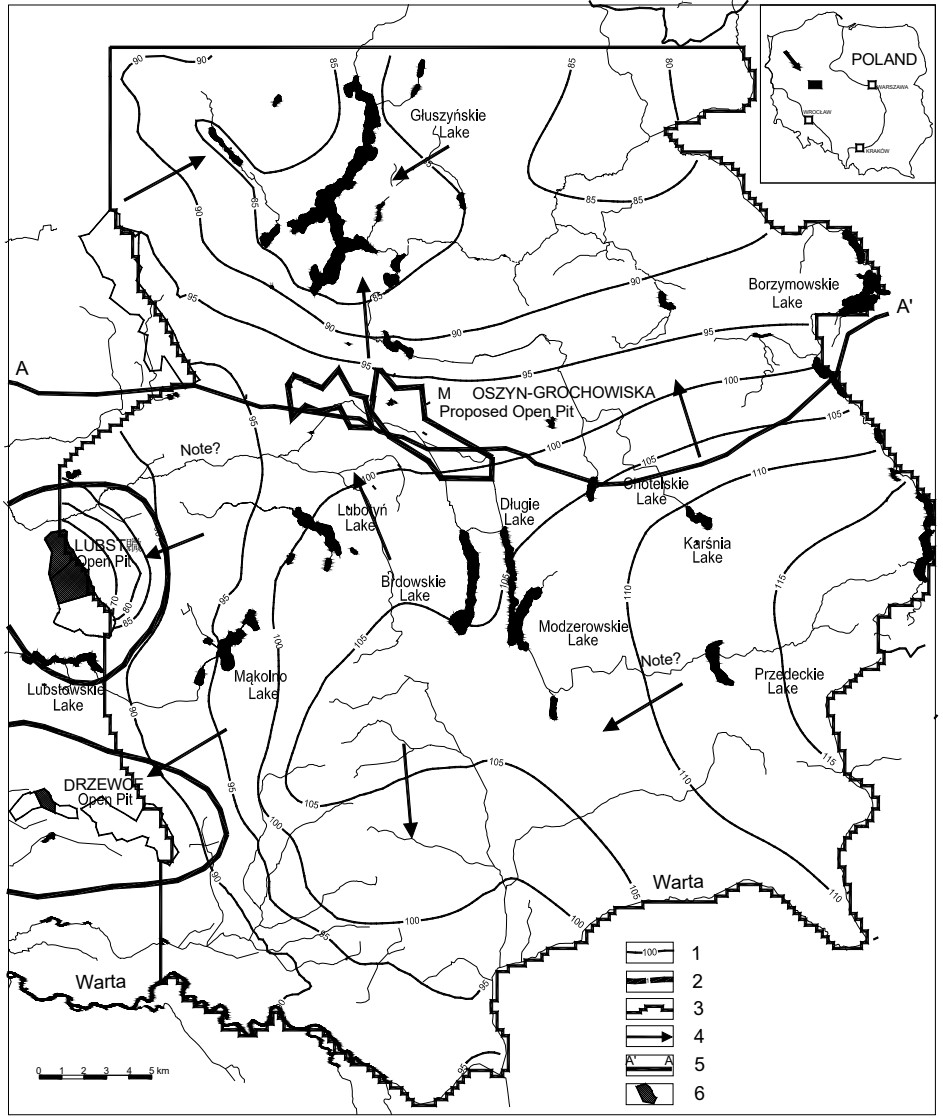

**Figure 7.** Groundwater flow modeling study for the proposed Mąkoszyn-Grochowiska open pit in the Konin lignite basin for pre-mining conditions. Explanations: **1**—groundwater level in pre-mining conditions; **2**—actual cone of depression range; **3**—model boundary; **4**—groundwater flow direction; **5**—hydrogeological cross section; **6**—mining areas [59].

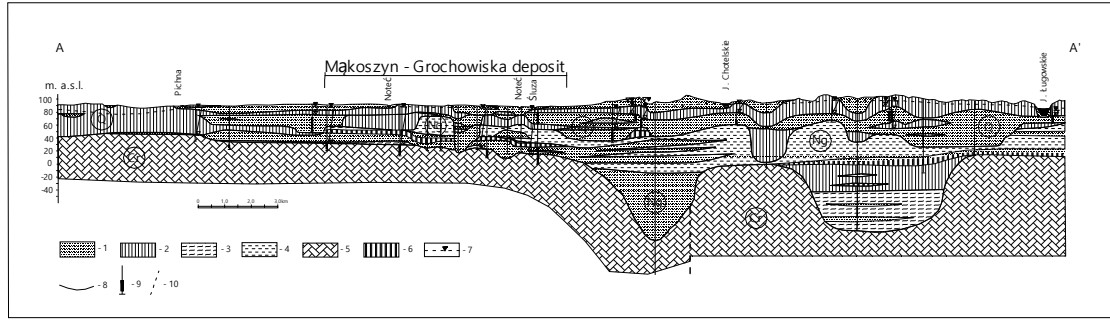

**Figure 8.** Hydrogeological cross-section through Mąkoszyn Grochowiska deposit. Explanations: **1**—fine sands; **2**—glacial tills; **3**—silts; **4**—clays; **5**—marls and dolomites; **6**—lignite; **7**—Neogene-Paleogene-Mesozoic aquifer piezometric surface; **8**—stratigraphic boundary; **9**—screened intervals; **10**—fault [59].

By the 1990s, the numerical models for the mining industry in Poland mainly solved the problem of the amount of mine water inflow and environmental impact assessment caused by mine water drainage. The problem of post-mining excavations flooding had not been extensively addressed since there were not many such reservoirs in Poland. Presently, the problem of filling post-mining voids with water is more extensively addressed for the reasons of estimating the costs of works related to flooding lignite opencast mines (Figures 9 and 10). It is foreseen that in the next years, the total volume of flooded post-mining voids in Polish lignite industry will reach more than 3.5 billion cubic meters (Table 1). The most crucial elements in modeling studies for water management of post-mining excavations are: The rate of flooding, restoration of the groundwater table, the impact of the reservoirs on groundwater, and the changes in the water quality in reservoir and aquifer.

**Table 1.** Post-mining voids in the Polish lignite industry.

| Opencast Mines | Deposit Exploitation | Number of Voids [1] | Area (km²) | Volume (mln m³) |
|---|---|---|---|---|
| Adamów | 1964–2018 | 8 | 10.6 | 252 |
| Konin | 1946–2030 | 10 | 21.8 | 664 |
| Bełchatów | 1981–2038 | 2 | 32.5 | 2422 |
| Turów | 1904–2040 | 1 | 17.0 | 1220 |

[1] Some after a rehabilitation.

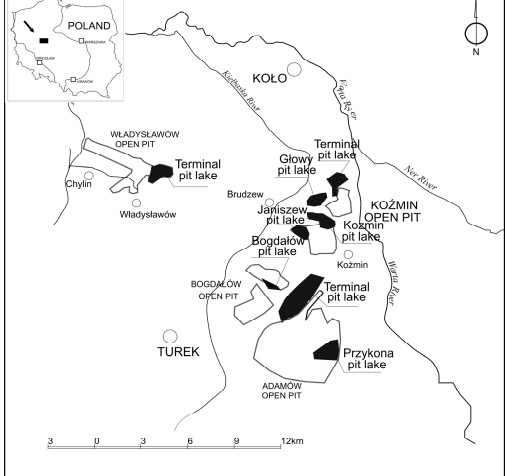
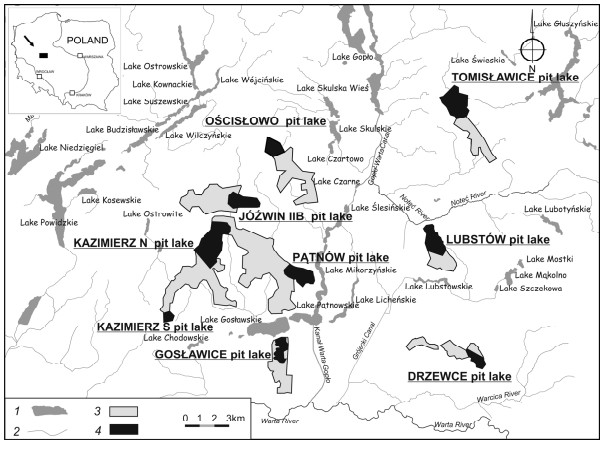

**Figure 9.** Current and proposed location of pit lakes in the Adamow (left) and the Konin post-mining area (right). Explanations: **1**—natural lake; **2**—river; **3**—post-mining area; **4**—pit lake [61].

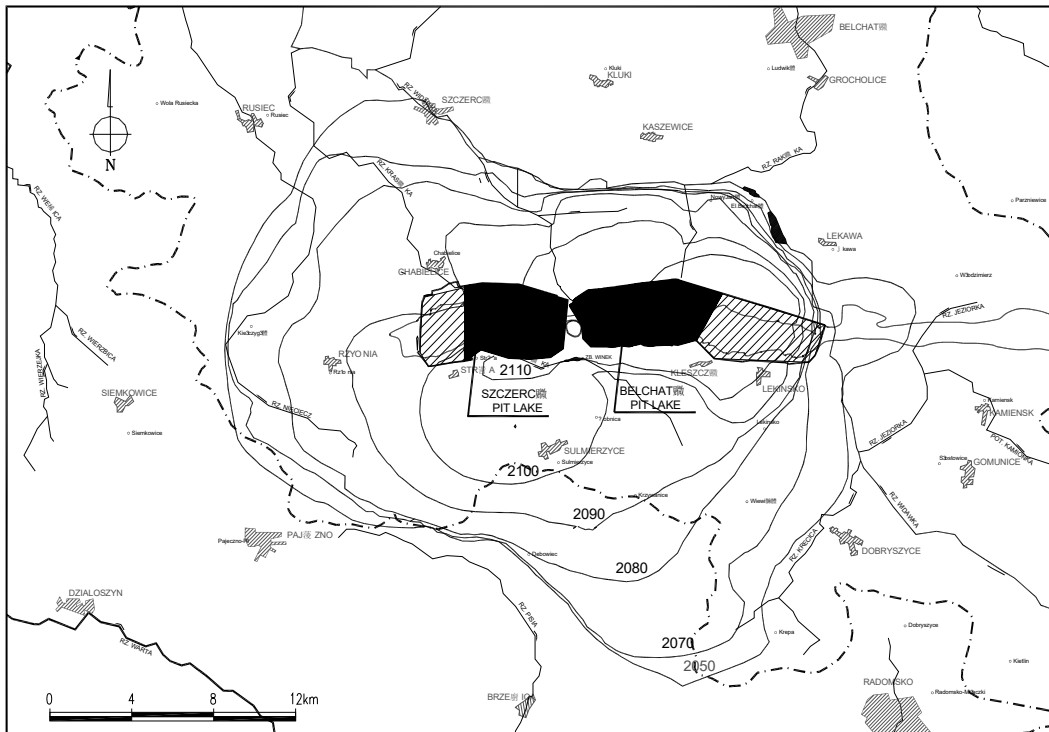

**Figure 10.** Proposed location of pit lakes in the Belchatow post-mining area with the cone of depression reduction, in case of no additional recharge [57].

## 6. Limitations of Modeling Methods

Numerical models used in mining hydrogeology, especially in opencast mining, have certain limitations and uncertainties related to the assumed stresses and parameters in the model. They include shortcomings and variability in the boundary conditions, hydrogeological and spatial parameters, groundwater recharge, mining schedules, and others. A regional model to analyze the effect of an open pit on groundwater levels often requires a lot more discretization to analyze the slope stability in the same pit [62]. The finer nodal spacing is required to define highly curved surfaces. It enables a simulation of a greater accuracy the decline in groundwater level and groundwater level itself near the slopes, which results in changes of geotechnical conditions of slopes in time and space as the mine develops.

The proper identification of hydraulic parameters, the nature of faults and fissures, as well as karstic formation phenomena, is decisive for the reliability of modeling studies. For example, natural hydraulic barriers, such as faults filled with semi-permeable material or rock formations with low hydraulic conductivity compared to adjacent rock layers, can decrease the mine water inflow and reduce the negative environmental impact of dewatering. However, while the deposit is under dewatering, new information based on the actual impact of the mine drainage on the groundwater and surface water may result in the necessity of adjusting hydraulic parameters of aquifers and semi-permeable layers in the model. Moreover, one should be aware that under the mine dewatering hydraulic parameters may change in time due to the activation of a new groundwater flow pathway.

Despite the fundamental importance of recharge in the water balance, it is an element with the highest uncertainty, since its values primarily depend on the accuracy assumed when calculating other elements of the water balance equation. Though it is one of the most important components in groundwater studies, recharge is also one of the least understood, largely because recharge rates vary widely in space and time, and rates are difficult to directly measure [38]. Changes in the recharge may be the consequence of many factors which were not accurately estimated or identified inside and outside the mining area. These may include: Changes of recharge caused by groundwater fluctuation

and land management, stream-aquifer interaction, unidentified leakage of surface waters into aquifers, water losses from the water supply and sewage disposal system, and industrial waters recharging the aquifers. Moreover, mining and power generation activity, including the construction of large open pits and the overburden of dumping areas, as well as the existence of powerful heat and steam emitters at nearby power plants, may lead to local climate variations, influencing recharge from precipitation.

Assuming the amount of aquifer recharge is based on average multiannual precipitation does not account for the cyclic deviations and trends resulting from climate variability. For instance, low precipitation may lead to a fall of the groundwater table below a level referred to as the evapotranspiration extinction depth, which causes an increase in effective infiltration. On the other hand, a change in the difference between potential evaporation and the precipitation affects the rate at which post-mining reservoirs are filled with water.

Several years after the modeling study is completed, a post audit is conducted. Its goal is to determine whether the prediction was correct, based on new field data. Most often, errors in model forecasts are not the result of imperfections in programs, but mistakes made by modelers. In general, a post-audit studies found that errors in model predictions were caused by errors in the conceptual model of the hydrogeological system [28] and a failure to use appropriate values for assumed future stresses such as recharge and pumping rates [63,64]. In a modeling study for the simulation of opencast mine activity, the different boundary conditions which are not compatible with the actual conditions are the most serious problem. Due to changes in the schedules of mining activities or rehabilitation of post-mining excavations, the input data used in the model should be verified and updated with reference to the most recent recommendations and assumptions.

## 7. Conclusions

Even though groundwater flow models used in mining hydrogeology have numerous limitations related to the uncertainty of the parameters and boundary conditions, they still provide the most comprehensive information concerning the mine dewatering system and its environmental impact at the time when they are developed. However, they will always require periodical verification based on new information on the actual response of the aquifer system to the mine drainage and the actual climate conditions, as well as up-to-date schedules of deposit extraction and mine closure. Numerical modeling used in mining hydrogeology needs a lot of experience, not only in hydrogeology but also in mining operations and other activities realted to mining industry in the neighboring areas.

**Funding:** This research received no external funding.

**Conflicts of Interest:** The author declares no conflict of interest.

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
