# Peer review of "The Significance of Groundwater Flow Modeling Study for Simulation of Opencast Mine Dewatering, Flooding, and the Environmental Impact"

_water, doi:10.3390/w11040848_

Round 1
Reviewer 1 Report
That the modeling with the program FeFlow is appropriate is shown in the article, which relates to the underground drainage of the coal.
Add to the row 92 The present state of groundwater flow modeling applications in mining, hydrogeology was presented by Rapantova et al. [16], and add to References.
VUKELIČ, Željko, DERVARIČ, Evgen, ŠPORIN, Jurij, VIŽINTIN, Goran. The development of dewatering predictions of the Velenje coalmine. Energies, ISSN 1996-1073, 2016, vol. 9, no.9, https://www.mdpi.com/1996-1073/9/9/702
Reviewer 2 Report
Please see all of my comments in the attached PDF.

Reviewer 3 Report
In my opinion, the studied subject of this paper is of high interest. The manuscript is clearly written and the methodology and example provided are well presented. Results obtained from mine dewatering modelling have a relevant weight for the considered field of application and the author clearly presented all the issues one could think of in the context of mine dewatering impact on the environment. Even if the novelty of this work is not significant, the guidance and experience sharing by the author is highly valuable and would benefit for any hydrogeologist. The paper is of high quality, easy to read and understand. Based on that, I think this paper should be accepted after minor revision, with of couple of spelling mistakes and corrections to be done in the text.
In addition, here are a non-exhaustive list of comments that should be addressed to improve this manuscript:
- Line 34: I suggest that you delete your sentence about the conference paper and rather cite it as you would do for a regular citation at some place in the text.
- Line 42: Please cite a reference textbook as an example.
- Line 45: “accuracy, than” should be “accuracy than”.
- Lines 77-78: which groundwater flow equation do you thin of? Please provide at least the name of it (Darcy, Richards’, etc.) and a citation to state it.
- Line 300: “of the this” should be “of this”.
- Line 301: “the seepage of water” should be “water seepage”.
- Figures 4, 5, 8 and 8 are very hard to read in black and white. You should add colors to these figures so that the lines and patches are easily distinguishable.
- Line 386: “Moreover, it should” should be “Moreover, one should”.
- Line 422: “The numerical modeling” should be “Numerical modeling”.
- Line 423: “hydrogeology need a” should be “hydrogeology needs a”.
